# Iron deficiency, anemia and association with refugee camp exposure among recently resettled refugees: A Canadian retrospective cohort study

Marta B. Davidson[1¤]*, Garielle Brown[2,3], Lesley Street[1], Kerry McBrien[2,4], Eric Norrie[2,3], Andrea Hull[3,4], Rachel Talavlikar[3,4], Linda Holdbrook[2], Gabriel E. Fabreau[2,3,5]

1 Division of Hematology and Hematologic Malignancies, Department of Medicine, Cumming School of Medicine, University of Calgary, Calgary, Canada, 2 Department of Community Health Sciences, O'Brien Institute for Public Health, Cumming School of Medicine, University of Calgary, Calgary, Canada, 3 Mosaic Refugee Health Clinic, Mosaic Primary Care Network, Calgary, Canada, 4 Department of Family Medicine, Cumming School of Medicine, University of Calgary, Calgary, Canada, 5 Division of General Internal Medicine, Department of Medicine, Cumming School of Medicine, University of Calgary, Calgary, Canada

¤ Current address: Division of Medical Oncology and Hematology, Department of Medicine, University Health Network, Princess Margaret Cancer Centre, Toronto, Ontario, Canada

* mbdavids@ucalgary.ca

**Data Availability Statement:** Data cannot be shared publicly because permission is restricted by the data custodian and current active research

## Abstract

Malnutrition and poor health are common among recently resettled refugees and may be differentially associated with pre-migration exposure to refugee camp versus non-camp dwelling. We aimed to investigate the associations of iron deficiency (ID), anemia, and ID anemia (IDA) with pre-migration refugee camp exposure among recently arrived refugees to Canada. To this end, we conducted a retrospective cohort study of 1032 adult refugees who received care between January 1, 2011, and December 31, 2015, within a specialized refugee health clinic in Calgary, Canada. We evaluated the prevalence, severity, and predictors of ID, anemia, and IDA, stratified by sex. Using multivariable logistic regression, we estimated the association of refugee camp exposure with these outcomes, adjusting for age, months in Canada prior to investigations, global region of origin, and parity. Among female refugees, the prevalence of ID, anemia, and IDA was 25% (134/534), 21% (110/534), and 14% (76/534), respectively; among males, 0.8% (4/494), 1.8% (9/494), and 0% (0/494), respectively. Anemia was mild, moderate, and severe in 55% (60/110), 44% (48/110) and 1.8% (2/110) of anemic females. Refugee camp exposure was not associated with ID, anemia, or IDA while age by year (ID OR = 0.96, 95% CI 0.93–0.98; anemia OR = 0.98, 95% CI 0.96–1.00; IDA OR = 0.96, 95% CI 0.94–0.99) and months in Canada prior to investigations (ID OR = 0.85, 95% CI 0.72–1.01; anemia OR = 0.81, 95% CI 0.67–0.97; IDA OR = 0.80, 95% CI 0.64–1.00) were inversely correlated with these outcomes. ID, anemia, and IDA are common among recently arrived refugee women irrespective of refugee camp exposure. Our findings suggest these outcomes likely improve after resettlement; however, given proportionally few refugees are resettled globally, likely millions of refugee women and girls are affected.

agreement. For further inquiries regarding the data sharing restrictions please contact the data custodian's (the Mosaic Primary Care Network) privacy officer Dr. Valerie Fleisch (privacy@mosaicpcn.ca).

**Funding:** GEF received research grant support from the MSI Foundation, the O'Brien Institute for Public Health and Department of Medicine at the University of Calgary for the submitted work

**Competing interests:** I have read the journal's policy and the authors of this manuscript have the following competing interests: GEF received research grant support from the MSI Foundation, the O'Brien Institute for Public Health and Department of Medicine at the University of Calgary for the submitted work. Otherwise, the authors declare no financial relationships or conflicts of interests with organizations that might have an interest in the submitted work in the previous three years; no other relationships or activities that could appear to have influenced the submitted work.

## Introduction

The world is facing an unprecedented global refugee crisis with an estimated 89.3 million people forcibly displaced from their homes and 27.1 million global refugees [1]. Refugees endure food and housing insecurity, limited healthcare access, and traumatic experiences that negatively impact their health status [2]. Currently, approximately 24% of the world's refugees reside in camps, and the remaining majority live in urban and rural regions within host-nations [3–5]. Further, high burdens of malnutrition and poor health are common among refugee camp inhabitants [6–8]. The United Nations High Commission for Refugees (UNHCR) now aims to limit camp deployment, instead encouraging integrating refugees within urban host-nation communities whenever possible [3]. It is unclear how these policy changes may affect the nutrition and health of refugees.

Worldwide, iron deficiency (ID) is the most common nutritional deficiency, and anemia is its most commonly associated adverse health outcome [9–14]. ID and iron deficiency anemia (IDA) are most commonly caused by inadequate access to iron-rich foods or poor iron absorption [15]. In women and adolescent girls, ID is also linked to menstrual blood loss and high parity [16]. Further, enteric pathogens from exposures to contaminated drinking water or foods can cause chronic intestinal blood loss leading to ID [17]. Globally, anemia has been linked to poor pregnancy outcomes [18, 19], reduced work productivity [20, 21], increased morbidity and increased all-cause mortality [22]. Refugees residing in refugee camps experience high prevalence of anemia and nutritional deficiencies [23], especially among women and children [10, 24–28]. It is currently unclear, however, if refugees in non-camp settings experience similar rates of nutritional deficiencies and anemia.

We aimed to characterize the prevalence and severity of ID, anemia, and IDA among recently-arrived adult refugees and claimants with and without pre-migration refugee camp exposure, and its differential association with these outcomes. We hypothesized that recently resettled refugees and asylum claimants exposed to refugee camps have higher prevalence and severity of ID, anemia, and IDA due to increased nutritional deficiencies compared to refugees living outside of refugee camps. Finally, we also aimed to characterize anemia unrelated to ID in this population.

## Methods

### Setting and data sources

We performed a retrospective cohort study among recently arrived refugees who presented to the *Mosaic Refugee Health Clinic* (MRHC) in Calgary, Canada. The MRHC is the only specialized refugee health clinic in its geographic area and provides multidisciplinary primary and specialty care to refugees and asylum claimants (claimants) for up to two years after resettlement. We performed a comprehensive chart review of electronic medical records (EMRs) using combined manual and automated methods to collect detailed sociodemographic, resettlement, clinical, biometric, laboratory, and clinic utilization data.

### Study population

We included all refugees and claimants, herein referred to collectively as "refugees", presenting to MRHC between January 1, 2011 and December 31, 2015 who: (1) were at least 18 years old at clinic intake, (2) had at least 2 clinic appointments, and (3) completed relevant laboratory investigations within 180 days of arrival to Canada, to investigate the associations of pre-migration refugee camp exposure with nutritional status and our outcomes of interest. We excluded subjects missing important data.

## Study variables

In Canada, convention refugees are defined as "a person who meets the refugee definition in the 1951 Geneva Convention relating to the Status of refugees [29], and are classified as either government assisted refugees (GAR) or privately sponsored refugees (PSR) [30]. We defined claimants as either subjects that were seeking protection in Canada but had not yet received convention refugee status, or those who had their refugee claims rejected during the study period. We extracted sociodemographic data including sex, date of birth, country of origin, refugee category, and date of arrival to Canada from EMR-recorded immigration documents. We categorized country of origin according to the United Nations' Geoscheme [31], and defined it as either a subject's country of birth, citizenship or inhabitation prior to migration, depending on where they lived the longest. We considered refugee camp exposure present if it was recorded in the intake history, regardless of duration. Two chart abstractors followed a standardized protocol to manually abstracted each study subject's clinical information from the EMR and categorize clinical diagnoses into ICD-10 codes [32]. Agreement between chart abstractors was >90% for clinical diagnostic codes; disagreements were resolved by a senior clinician. Among female subjects, pregnancy status and number of live children were recorded at intake. We considered subjects pregnant if pregnancy was confirmed at the time of laboratory investigations and assumed parity to be zero if it was not explicitly recorded in the EMR.

## Study outcomes

We analyzed laboratory data to identify our primary outcomes of interest: ID, anemia, and IDA. Anemia was defined according to World Health Organization (WHO) definitions as: a hemoglobin <120g/L for all non-pregnant females ≥15 years of age, <110g/L for pregnant females ≥15 years of age, and <130g/L for males ≥15 years of age [33]. Among anemic refugees, we assessed anemia severity and morphology. We defined mild, moderate, and severe anemia among non-pregnant women as a hemoglobin concentration of: 110-119g/L, 80-109g/L, and <80g/L, respectively, and among pregnant women as 100-109g/L, 70-99g/L, and <70g/L, respectively [33]. In adult males ≥18 years of age, we defined mild, moderate, and severe anemia as a hemoglobin concentration of 110-129g/L, 80-109g/L, and <80g/L, respectively [33]. For all subjects we further classified anemia as: microcytic (<77 femtolitres [fl]), normocytic (77–96fl), or macrocytic (>96fl) [34]. Finally, we defined iron deficiency (ID) according to WHO standards as a serum ferritin concentration of <15μg/L [35, 36] and IDA as the concomitant presence of anemia and ID as outlined above.

## Missing data

Fifty-six individuals were missing data for "number of children", which was replaced with the most frequent and median number of children: two. We tested this assumption by replacing these 56 missing data points with zero, which did not change any study outcomes.

## Statistical analysis

We stratified our analysis by sex given the known variation in prevalence of iron deficiency and anemia with sex [24]. We reported descriptive statistics according to refugee camp exposure, stratified by sex. We reported normally-distributed continuous variables as means with standard deviations (SD), and all non-normally distributed variables as medians with interquartile ranges [IQR]. We assessed statistical differences among demographic factors by refugee camp exposure using the Wilcoxon rank sum tests for non-normally-distributed variables, and the chi-square or Fisher's exact tests for normally distributed variables.

We used univariate logistic regression models to estimate unadjusted odds ratios for the outcomes of interest, comparing refugees with and without camp exposure. Given the very low prevalence of anemia among males, we restricted our adjusted analyses to women only.

We initially utilized multivariable, two-level, logistic regression models with a random effect for UN region of origin, to account for non-random clustering of patients according to global regions and their associations with our outcomes of interest [19]. We allowed the UN region intercepts to vary randomly among patients (level 1) within similar UN global regions of origin (level 2). We then investigated the degree of patient clustering within global regions and sub-regions and its association with the variances observed for our outcomes of interest. The intraclass correlation coefficients was 0.001 for all three outcomes, indicating a negligible degree of clustering within global regions and sub-regions thus negating the need for multi-level regression models. We thus used multivariable logistic regression models to estimate the adjusted odds for the outcomes of interest among female refugees adjusting for refugee camp exposure, age, UN global region of origin (Africa as reference group), number of months in Canada prior to laboratory investigations, active pregnancy status (for ID), and number of children (parity).

### Sensitivity analysis

To test our definition of ID based on serum ferritin concentration, we performed a sensitivity analysis whereby we assessed our outcomes of interest using ID defined as a transferrin saturation (tsat) of less than 0.16 [33] and IDA as anemia with a concomitant tsat <0.16.

### Exploratory analysis

For all anemia cases not associated with ID (ferritin <15g/L), we explored patient records for alternate etiologies of anemia and characterized these conditions by ICD-10 diagnostic codes. We then classified these conditions into the following categories: B12 deficiency, thalassemic conditions, ID, and other if patients did not have any anemia-specific ICD-10 codes documented within 6 months of arrival to Canada. ID or IDA diagnoses were excluded if the patient's ferritin concentration was above 100g/L, a level largely considered adequate to rule out ID even in context of inflammatory states [37]. In these cases, we sought other anemia-diagnoses from patients' charts.

All analyses were performed using Stata 14 (College Station, TX USA: StataCorp LLC) and SAS 9.4 (Cary, NC, USA: SAS Institute Inc.) and considered a 2-tailed alpha <0.05 as statistically significant. This study was approved by the University of Calgary Research Ethics Board.

## Results

### Baseline characteristics

We identified 1032 adult refugee patients in our cohort, with 52% (534/1032) females and 48% (498/1032) males (Fig 1). The median age was 32.6 years [26.3–41.3] and 31.2 years [25.2–39.8] among males and females, respectively. Subjects originated from 47 different countries. The top five countries of origin were Eritrea (23%), Iraq (14%), Ethiopia (13%), Afghanistan (6%) and Bhutan (6%). Among UN global regions, 53% (542/1032) of refugee patients were from Africa, 45% (461/1032) were from Asia, 2% (22/1032) were from the Americas, and 1% (10/1032) were from Europe (Table 1). Overall, 18.7% (193/1032) of refugees were exposed to a refugee camp, with African and government assisted refugees (GAR) overrepresented among those exposed. Among females, English proficiency varied by refugee camp exposure, but pregnancy rates and number of children did not (Table 1).

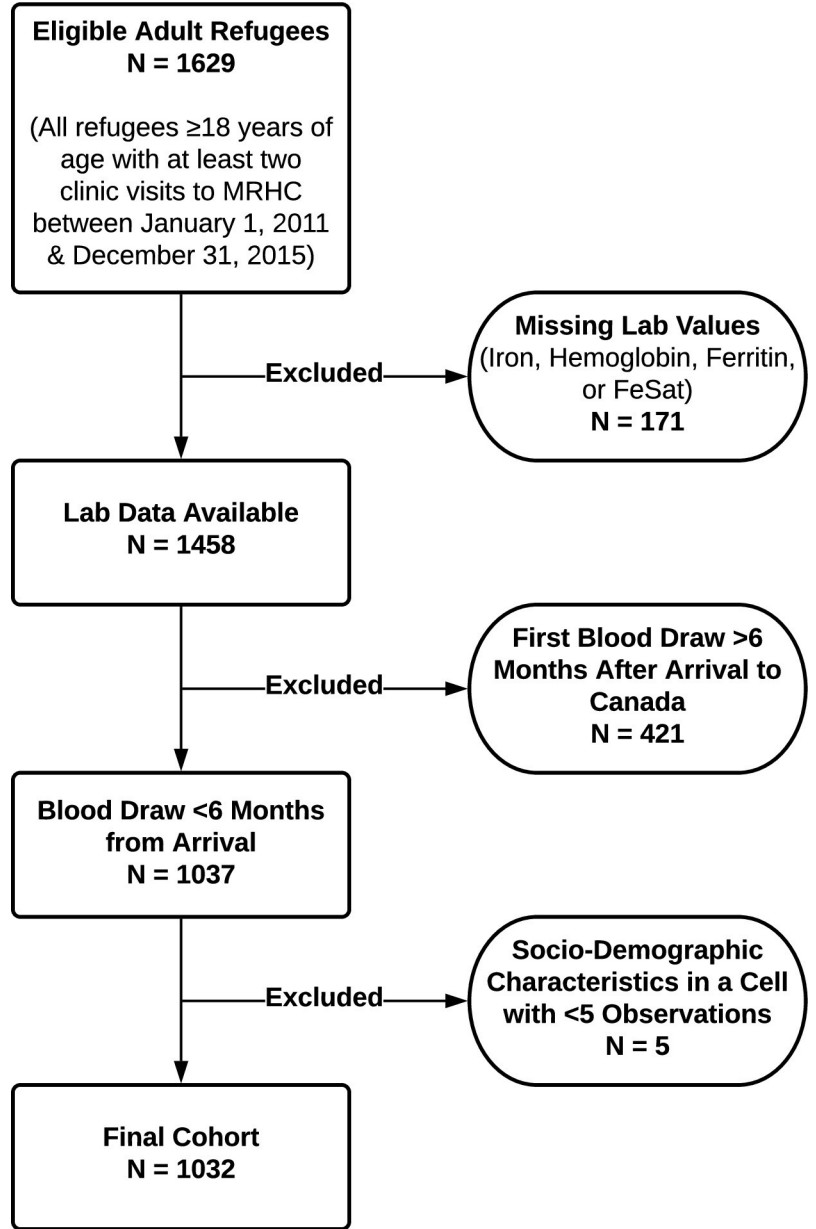

**Fig 1. Flow chart of study.** See text for details.

### Prevalence of ID, anemia, and IDA among refugees

Among all subjects, the median time from arrival to Canada to first laboratory investigations was 39 days [22–72 days]. Among females and males, the prevalence of ID was 25% (134/534) and 0.8% (4/498), respectively (p<0.01) (Fig 2A), the median ferritin level was 29μg/L [14–57μg/L] and 120.5μg/L [70–186μg/L] (p<0.01), respectively, and the prevalence of anemia was 21% (110/534) and 1.8% (9/498), respectively (p<0.01). Sixty-nine percent (76/110) of anemia among females was attributed to ID. There were no IDA cases among males. Among anemic females, 55% (60/110) had mild, 44% (48/110) had moderate, and 2% (2/110) had severe anemia. Among anemic males, 78% (7/9) had mild, 22% (2/9) had moderate, and none had severe

**Table 1. Patient characteristics by sex and refugee camp exposure (N = 1032).**

| Variable | Female N = 534 | | | Male N = 498 | | |
|---|---|---|---|---|---|---|
| | Refugee Camp +<br>N = 94 | Refugee Camp -<br>N = 440 | p-value* | Refugee Camp +<br>N = 99 | Refugee Camp -<br>N = 399 | p-value* |
| **Age** (years)–Median [IQR] | 30.3 [23.6–38.3] | 31.4 [25.5–40.0] | 0.28 | 32.0 [25.5–42.4] | 32.7 [26.5–41.1] | 0.92 |
| **Months in Canada Prior to Blood Tests**–Median [IQR] | 1.1 [0.7–1.9] | 1.4 [0.8–2.5] | 0.08 | 1.2 [0.8–2.1] | 1.3 [0.7–2.6] | 0.69 |
| **UN Global Region**–N (%) | | | | | | |
| Africa | 63 (67.0) | 206 (46.8) | < 0.01 | 69 (69.7) | 204 (51.1) | < 0.01 |
| Americas | 0 (0) | 17 (3.9) | | 0 (0) | 5 (1.3) | |
| Asia | 31 (33) | 210 (47.7) | | 30 (30.3) | 190 (47.6) | |
| Europe | 0 (0) | 7 (1.6) | | 0 (0) | 0 (0) | |
| **Number of Children**–Median [IQR] | 1 [0–4] | 2 [0–3] | 0.43 | 1 [0–2] | 1 [0–2] | 0.86 |
| **Pregnancy^**–N (%) | | | | | | |
| Yes | 6 (6.4) | 48 (10.9) | 0.19 | | N/A | |
| No | 88 (93.6) | 392 (89.1) | | | | |
| **Refugee Category#** | | | | | | |
| GAR | 71 (75.5) | 204 (46.4) | < 0.01 | 65 (65.7) | 188 (47.1) | < 0.01 |
| PSR | 23 (24.5) | 191 (43.4) | | 34 (34.3) | 180 (45.1) | |
| Claimant | 0 (0) | 45 (10.2) | | 0 (0) | 31 (7.8) | |
| **English Language proficiency at intake**–N (%) | | | | | | |
| None | 61 (64.9) | 214 (48.6) | | 38 (38.4) | 129 (32.3) | |
| Limited | 11 (11.7) | 73 (16.6) | | 20 (20.2) | 86 (21.6) | |
| Good | 22 (23.4) | 153 (34.8) | 0.02 | 41 (41.4) | 184 (46.1) | 0.52 |

*p-values calculated using a Wilcoxon rank sum test for comparison of medians and chi-square or Fisher's exact tests for comparison of frequencies.

#GAR refer to Government assisted refugees, PSR refers to privately sponsored refugees, and Claimant refers to refugee claimants or asylum seeker.

^Pregnant refers to active pregnancy at the time of laboratory investigations. N/A refers to not applicable.

anemia (Fig 2B). Among anemic subjects, most anemia was normocytic and only one female had macrocytic anemia (Fig 2C).

**Unadjusted outcomes.** Table 2 summarizes the prevalence of ID, anemia, and IDA among all subjects by refugee camp exposure stratified by sex. There were no prevalence differences in the outcomes of interest by refugee camp exposure. In our unadjusted analysis restricted to females (Table 3), refugee camp exposure, UN region of origin, and number of children were not associated with any of the outcomes of interest; however, each increased year of age and each month in Canada prior to laboratory investigations were inversely associated with these outcomes. Pregnancy was associated with ID but was not tested against anemia or IDA as we used pregnancy specific definitions for these outcomes (Table 3).

## Adjusted analysis

In our adjusted analysis, refugee camp exposure was not associated with our outcomes of interest among female refugees (Fig 3). Further, UN region of origin, pregnancy, and number of children were also not associated with ID, anemia, or IDA. Each increased year of age at first clinic appointment was associated with 4% and 3% decreased odds of ID and IDA, respectively (Fig 3). Finally, each month in Canada prior to when laboratory investigations were drawn was associated with 19% and 20% decreased odds of anemia and IDA, respectively (Fig 3).

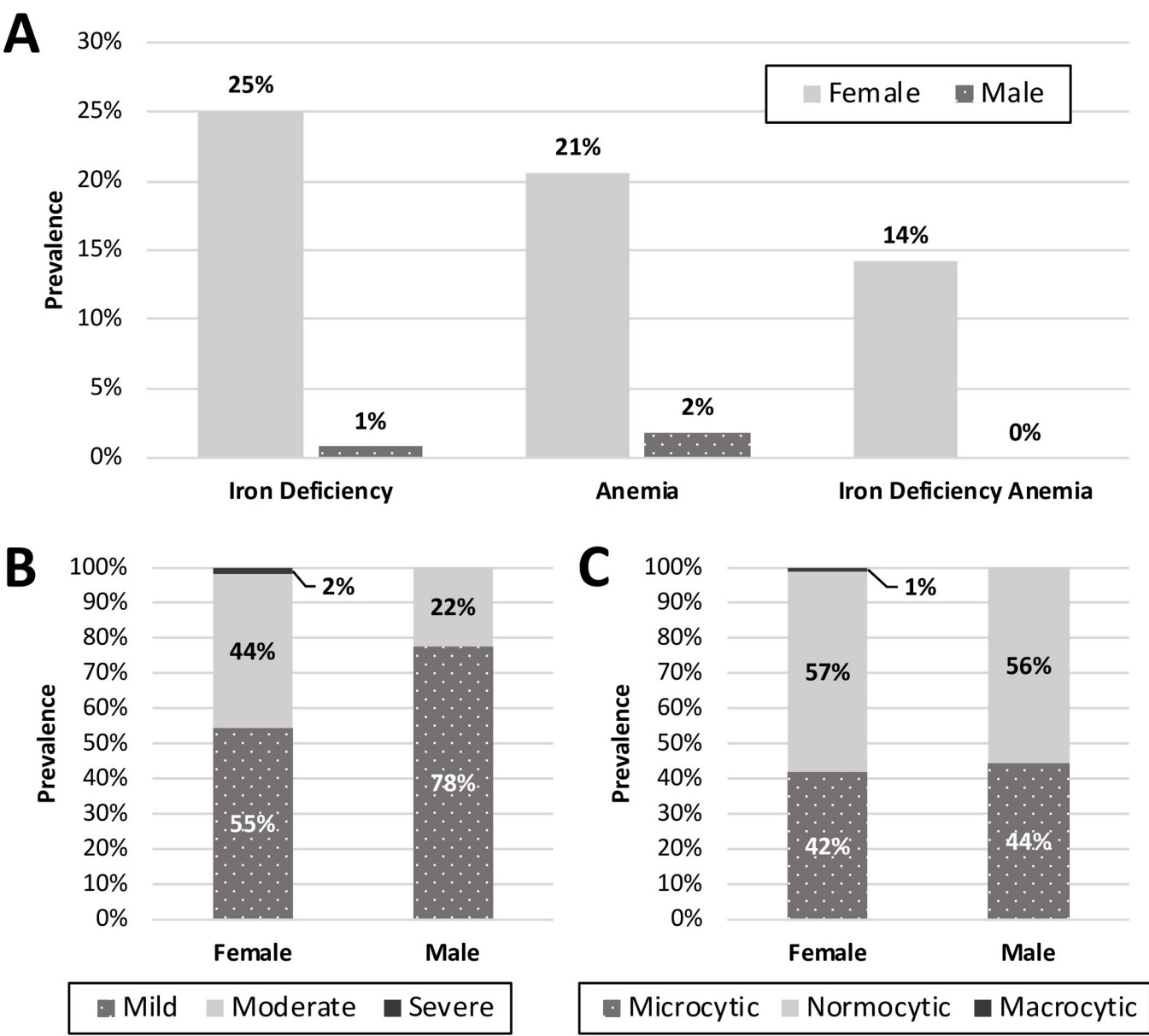

**Fig 2. Iron deficiency and anemia characteristics in adult refugees resettled to Calgary, Alberta.** (A) Prevalence of ID, anemia, and IDA in male and female refugees. We defined ID as a serum ferritin <15μg/L and anemia according to the WHO standards [15]. (B) Anemia severity in male and female refugees. Among non-pregnant women we defined mild, moderate, and severe anemia as a hemoglobin concentration of 110-119g/L, 80-109g/L, and below 80g/L, respectively, and 100-109g/L, 70-99g/L, and <70g/L, respectively among pregnant women. In adult males ≥18 years of age, we defined mild, moderate, and severe anemia as hemoglobin concentrations of 110-129g/L, 80-109g/L, and <80g/L respectively. (C) Anemia morphology. Anemia was classified as microcytic (<77fl), normocytic (77–96fl), or macrocytic (>96fl).

## Sensitivity analysis

In our sensitivity analysis, the overall prevalence of ID obtained using transferrin saturation (tsat) [21.5% (222/1032)] was higher than using a ferritin-based definition [13.3% (138/1032)] (S1 Table). However, this did not change the results of our unadjusted (S2 Table) or adjusted analysis (S1 Fig).

**Table 2. Primary outcomes by sex and refugee camp exposure (N = 1032).**

| Variable | Female N = 534 | | | Male N = 498 | | |
|---|---|---|---|---|---|---|
| | Refugee Camp + | Refugee Camp - | p-value* | Refugee Camp + | Refugee Camp - | p-value* |
| | N = 94 | N = 440 | | N = 99 | N = 399 | |
| Iron Deficiency–N (%) | 19 (20.2) | 115 (26.1) | 0.24 | 0 (0) | 4 (1.0) | 0.99 |
| Anemia–N (%) | 19 (20.2) | 91 (20.7) | 0.99 | 4 (4.0) | 5 (1.3) | 0.08 |
| Iron Deficiency Anemia–N (%) | 12 (12.8) | 64 (14.6) | 0.75 | 0 (0) | 0 (0) | – |

## Exploratory analysis: Determinants of non-iron deficiency associated anemia

Among anemia cases, 36% (43/119) were not attributable to ID as defined in our study. In these 43 non-ID subjects, physicians assigned diagnoses of IDA in 44% (19/43), thalassemia trait in 16% (7/43), and B12 deficiency in 5% (2/43) of individuals (Fig 4). Thirty-five percent (15/43) of these individuals did not have an explicit cause of anemia identified (Fig 4). Among patients with physician-assigned diagnoses of IDA, the median ferritin was 22.5μg/L [17–33μg/L]. Two patients were misclassified as having IDA by physician-assigned diagnoses as each had ferritin concentrations of >100μg/L; we therefore excluded these cases from median ferritin calculations. When we combined physician assigned cases of IDA with laboratory-defined cases, ID accounted for 80% (95/119) of anemia cases in our cohort.

## Discussion

Among a global cohort of adult refugees recently arrived in Canada, iron deficiency (ID), anemia, and iron deficiency anemia (IDA) were highly prevalent conditions irrespective of pre-migration refugee camp exposure. This was especially notable among female refugees, in whom the prevalence of ID was almost three times higher than that reported for adult Canadian females [38]. Contrary to our hypothesis, ID, anemia, and IDA were not associated with previous refugee camp exposure. However, we found that 82% of anemia cases among adult refugees were attributable to nutritional deficiencies (Iron or B12), and that inherited red blood cell disorders or other causes of anemia were rare. Given that refugees in our cohort completed laboratory investigations with a median time of 39 days after arrival to Canada, our

**Table 3. Unadjusted outcomes for females (N = 534).**

| Variables | Iron Deficiency | Anemia | Iron Deficiency Anemia |
|---|---|---|---|
| | Unadjusted Odds Ratio [95% CI] | Unadjusted Odds Ratio [95% CI] | Unadjusted Odds Ratio [95% CI] |
| Refugee Camp Exposure | 0.72 [0.41–1.24] | 0.97 [0.56–1.69] | 0.86 [0.44–1.67] |
| Age (in years) at First Appointment | **0.97 [0.95–0.99]** | **0.98 [0.96–1.00]*** | **0.96 [0.94–0.99]** |
| Months in Canada Prior to Blood Tests | **0.83 [0.71–0.97]** | **0.83 [0.70–0.99]** | **0.82 [0.66–1.00]**** |
| UN Global Region | Calculated vs. Africa | Calculated vs. Africa | Calculated vs. Africa |
| Americas | 0.75 [0.21–2.68] | 0.47 [0.11–2.13] | 0.38 [0.05–2.95] |
| Asia | 1.37 [0.92–2.05] | 0.86 [0.56–1.33] | 1.03 [0.63–1.70] |
| Europe | 2.61 [0.57–12.00] | 1.42 [0.27–7.53] | 2.43 [0.46–12.99] |
| Number of Children | 0.99 [0.90–1.10] | 0.93 [0.83–1.04] | 0.91 [0.79–1.04] |
| Pregnant^ | **1.89 [1.05–3.41]** | N/A | N/A |

*Results statistically significant, p = 0.012.

**Results statistically significant, p = 0.04.

^Pregnant refers to active pregnancy at the time laboratory investigations were drawn.

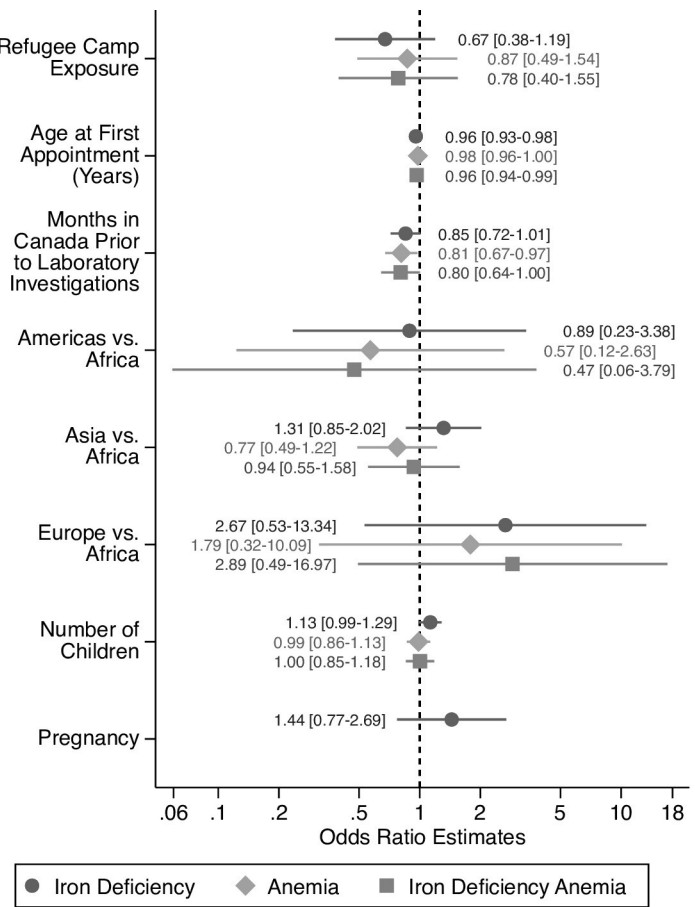

**Fig 3. Predictors of ID, Anemia, and IDA in recently resettled global female refugees.** Multi-variable logistic regression was used to estimate the odds of ID, anemia, and IDA in female refugees after adjusting for refugee camp exposure, age, global region of origin (defined by UN global regions), pregnancy status and number of children. (*) indicates p-value <0.05.

findings likely reflect pre-migration hematologic indices, as well as nutritional and health status.

The 21% anemia prevalence among refugee women in our study is consistent with previous reports [10, 25, 26, 39], and indicates a condition of moderate public health significance by WHO criteria [15]. We found female refugees had a 31-fold higher prevalence of ID and an 11-fold higher prevalence of anemia compared to males [24], consistent with higher anemia frequencies among women reported globally [10, 25, 26, 39]. We found that 25% of refugee women were iron deficient, which is likely an underestimate as we used a stringent ferritin concentration threshold of 15μg/L to define ID that likely misses milder, but clinically relevant deficiencies [40, 41]. Using our stringent ID definition, we found that almost 70% of anemic females had laboratory confirmed ID. Moreover, when we included physician assigned IDA cases with a median ferritin concentration of 22.5μg/L, ID accounted for 84% of observed anemia among women. This further suggests that our estimates are conservative, and that refugee women are at high risk for ID and anemia and their associated adverse health outcomes, irrespective of refugee camp exposure or global region or origin.

To our knowledge, this is the first study to investigate the association of nutritional and health status with pre-migration refugee camp exposure among global refugees, from 47

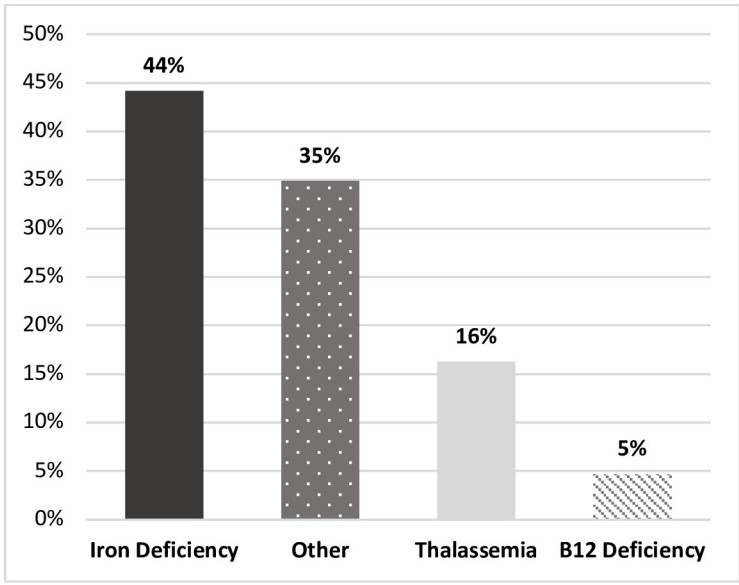

**Fig 4. Etiology of anemia not attributable to iron deficiency.** Conditions causing anemia using ICD10 diagnostic codes extracted from charts among patients with anemia, but a serum ferritin concentration >15µg/L (N = 43).

different countries, resettled to a high-income country. Overall, we found no differences in the prevalence of ID, anemia, and IDA among refugees by refugee camp exposure. Our findings suggest that refugees in non-camp settings (often in urban settings) may experience similar nutritional deficiencies as well as barriers accessing food and nutritional supports as camp-dwelling refugees. A recent systematic review also showed that nutritional deficiencies are a significant problem among non-camp dwelling refugees [42]. A number of factors may be contributing to this problem. For example, non-camp refugees may lack access to supports from the World Food Programme [43] and other international aid organizations that traditionally support refugees in camp settings. Further, the majority of displaced persons are relocated to lower income countries [1, 42], which likely have limited resources to offer to urban-dwelling refugees. Our findings appear to contradict the purported health and economic benefits of non-camp dwelling for refugees cited by the UNHCR such as greater freedom of movement and access to employment, given that nutritional deficiencies and food access barriers are most often related to poverty [3].

While high rates of ID and anemia have been documented in selected refugee camps [23], refugee camp exposure has been variably associated with anemia in the literature. For example, among pregnant Sudanese refugees in Ethiopia, anemia was inversely associated with duration of residence in a refugee camp [44]. By contrast, Syrian refugee women in refugee camps in Jordan, but not in Lebanon, were more likely to be anemic than their non-camp counterparts [6, 45]. In Southeast Asia, among refugees from Vietnam, Laos, and Cambodia, there was no association between anemia and refugee camp residence after accounting for ethnicity [19]. These observed differences may be due to the geographic and cultural variability between different refugee camps, as well as significant differences in their availability of nutrition supports and medical services. Unlike previous studies that have been limited to specific global sub-regions and individual ethnocultural populations, our study investigated a global cohort of refugees across 47 countries of origin. We found negligible clustering by UN global region of origin for ID, anemia, and IDA, suggesting that the associations of interest were not significantly different across global regions. Together our findings suggest that ID, often related to food

insecurity, is highly prevalent in pre-menopausal refugee women irrespective of pre-migration living situation or global origin.

Our study has limitations that warrant consideration. As our study was conducted at a single site, it may not be representative of refugee populations in other host-regions. Further, our study enrolment ended prior to the influx of approximately 41,000 Syrian refugees to Canada in 2015–2016, and more recent Afghan and Ukrainian refugees [46, 47], thus, it may not represent the current global refugee population. While our study is amongst the largest Canadian refugee cohort studies conducted to-date and comprised a heterogeneous global refugee cohort from 47 different countries, it is underpowered to assess country-specific variations which may factor into the lack of significant regional or sub-regional variation in our outcomes. Moreover, we only identified two cases of severe anemia in our study and may underestimate the prevalence of severe anemia cases that may present to acute care settings. This may be due to a selection bias given that our study relied on outpatient laboratory tests. Alternatively, selection pressures during the refugee migration process may select against individuals with severe anemia who may succumb to its complications or be unable to travel [48]. In addition, because our study was restricted to adult refugees and claimants, post-menarche adolescent girls, who likely also experience high prevalence of ID, anemia, and IDA were not included, thus potentially underestimating their prevalence. Finally, in our cohort, 19% of refugees had resided in a refugee camp prior to arrival in Canada, lower than current global trends, where 24% of global refugees reside in camps [4], suggesting camp-exposure may be underrepresented in our cohort.

Taken together, our findings, in concordance with other global data, suggest that refugee women commonly experience ID and IDA, and that refugee status itself, irrespective of pre-migration dwelling, region of origin, or political refugee class predicts these nutritional deficiency-related conditions. Our findings support the current *Canadian Collaboration for Immigrant and Refugee Health* (CCIRH) screening guidelines, which recommend screening all female refugees, but not males for anemia and ID [49]. In our cohort, anemia varied inversely with months spent in Canada prior to laboratory investigations, suggesting that it may be largely remediable in food-secure environments and with access to high quality primary healthcare.

The unprecedented number of forcibly displaced people globally highlights the scale of the population at risk for nutritional deficiencies. Further, the magnitude of the burden of ID and IDA among displaced women and adolescent females likely reaches at least a moderate public health concern by WHO criteria and warrants consideration of large-scale action to prevent adverse outcomes of this readily preventable nutritional deficiency. Globally, less than 2% of refugees are resettled in high income countries such as Canada; however, studies among multi-national cohorts of recently resettled refugees can improve understanding of health conditions faced by refugees regionally and universally [50]. Future studies of ID and anemia among global refugees would help to elucidate if regional and cultural differences exist, and if their prevalence varies across different nutritional and medical assistance models between refugee camps. Improved understanding of nutritional deficiencies in refugee populations may inform treatment and prevention programs, such as targeted iron supplementation programs for pre-menopausal refugee women and adolescent girls.

## Supporting information

**S1 Fig. Sensitivity analysis.** For sensitivity analysis, ID was defined as a serum transferrin saturation (tsat) $\leq 0.16$ and IDA was defined as the presence of anemia as defined by WHO thresholds in combination with a tsat $\leq 0.16$. Multi-variable logistic regression was used to

estimate the odds of ID, anemia, and IDA in female refugees after adjusting for refugee camp exposure, age, global region of origin (defined by UN global regions), pregnancy status and number of children.
(TIF)

**S1 Table. Sensitivity analysis; primary outcomes by sex and refugee camp exposure using serum transferrin saturation (tsat) (N = 1032).**
(DOCX)

**S2 Table. Sensitivity analysis: Unadjusted outcomes among female refugees using transferring saturation (tsat) (N = 534).**
(DOCX)

## Acknowledgments

We would like to thank the healthcare providers, staff and patients at the Mosaic Refugee Health Clinic the institutional support provided by, the O'Brien Institute for Public Health, the Department of Medicine and the W21C Research and Innovation Centre at the University of Calgary Cumming School of Medicine.

## Author Contributions

**Conceptualization:** Marta B. Davidson, Lesley Street, Kerry McBrien, Rachel Talavlikar, Gabriel E. Fabreau.

**Data curation:** Eric Norrie.

**Formal analysis:** Marta B. Davidson, Garielle Brown, Gabriel E. Fabreau.

**Funding acquisition:** Gabriel E. Fabreau.

**Investigation:** Andrea Hull, Rachel Talavlikar, Gabriel E. Fabreau.

**Methodology:** Marta B. Davidson, Gabriel E. Fabreau.

**Resources:** Gabriel E. Fabreau.

**Supervision:** Gabriel E. Fabreau.

**Writing – original draft:** Marta B. Davidson, Gabriel E. Fabreau.

**Writing – review & editing:** Marta B. Davidson, Garielle Brown, Lesley Street, Kerry McBrien, Eric Norrie, Andrea Hull, Rachel Talavlikar, Linda Holdbrook, Gabriel E. Fabreau.

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
