## [Editor Report · Decision Letter 0]

8 Jun 2021

PONE-D-21-12986

Iron deficiency, Anemia and Association with Refugee Camp Exposure Among Recently Resettled Refugees:  a Canadian retrospective cohort study

PLOS ONE

Dear Dr. Davidson,

Thank you for submitting your manuscript to PLOS ONE. After careful consideration, we feel that it has merit but does not fully meet PLOS ONE’s publication criteria as it currently stands. Therefore, we invite you to submit a revised version of the manuscript that addresses the points raised during the review process.

We look forward to receiving your revised manuscript.

Kind regards,

Sabeena Jalal, MBBS, MSc, MSc, SM

Academic Editor

PLOS ONE
---

## [Author Response · Author response to Decision Letter 0]

20 Sep 2021

Dear Editor:

Thank you for taking the time to review our manuscript entitled “Iron deficiency, anemia and association with refugee camp exposure among recently resettled refugees: a Canadian retrospective cohort study”, and for your insightful suggestions on how to improve it. 

We have addressed your requirements as follows:

1. “Please ensure that your manuscript meets PLOS ONE's style requirements, including those for file naming”

• You will now find the manuscript is in compliance with PLOS One style requirements. A marked-up copy and a “clean” copy of the manuscript have been uploaded.

2. “Please include captions for your Supporting Information files at the end of your manuscript, and update any in-text citations to match accordingly”

• Captions for Supporting Information files have now been included at the end of the manuscript and in-text citations have been updated accordingly.

3. “We note that you have indicated that data from this study are available upon request. PLOS only allows data to be available upon request if there are legal or ethical restrictions on sharing data publicly.”

• While we agree that sharing open data would enhance confidence and accountability in research, our current data-sharing agreement with the data custodian (The Mosaic Primary Care Network) from whom our clinical data comes, does not allow any sharing of data with third parties, even fully de-identified data. Since these requested revisions, I have reviewed our agreements with their privacy officer and have tried to advocate for sharing with your journal, but this would require a fully executed revision to our existing legal agreement which would take many weeks between their organization and the University of Calgary. Unfortunately, we would not be able to pursue these amendments and provide revisions in a reasonable amount of time to the journal. 

Despite this restriction, we have assembled and provided all our raw data output from our analyses, including the STATA code we developed to perform these analyses. We hope that providing our raw analyses will provide the reviewers confidence in our methodology and the results we present. We reviewed provision of these analyses and code with our research ethics board and data custodian, both of whom had no concerns with providing them for the journal. 

For further inquiries regarding the data sharing restrictions please contact the data custodian’s privacy officer Dr. Valerie Fleisch (privacy@mosaicpcn.ca).

4. “Please review your reference list to ensure that it is complete and correct. If you have cited papers that have been retracted, please include the rationale for doing so in the manuscript text, or remove these references and replace them with relevant current references. Any changes to the reference list should be mentioned in the rebuttal letter that accompanies your revised manuscript. If you need to cite a retracted article, indicate the article’s retracted status in the References list and also include a citation and full reference for the retraction notice.”

• The reference list has been reviewed and is complete and correct. There are no references to retracted articles.

We hope that you will find our revisions satisfactory. 

Marta Davidson

---

## [Decision Letter · Decision Letter 1]

31 Aug 2022

PONE-D-21-12986R1Iron deficiency, Anemia and Association with Refugee Camp Exposure Among Recently Resettled Refugees:  a Canadian retrospective cohort studyPLOS ONE

Dear Dr. Davidson,

Thank you for submitting your manuscript to PLOS ONE. After careful consideration, we feel that it has merit but does not fully meet PLOS ONE’s publication criteria as it currently stands. Therefore, we invite you to submit a revised version of the manuscript that addresses the points raised during the review process.

Thank you for  submitting this manuscript, and the revisions.  Please replace the pie chart by a bar graph. Please elaborate the points raised by one of the reviewers in your limitations section. Thank you.

We look forward to receiving your revised manuscript.

Kind regards,

Sabeena Jalal, MBBS, MSc, MSc, SM

Academic Editor

PLOS ONE

Journal Requirements:

Reviewers' comments:

Reviewer's Responses to Questions

**Comments to the Author**

1. If the authors have adequately addressed your comments raised in a previous round of review and you feel that this manuscript is now acceptable for publication, you may indicate that here to bypass the “Comments to the Author” section, enter your conflict of interest statement in the “Confidential to Editor” section, and submit your "Accept" recommendation.

Reviewer #1: All comments have been addressed

Reviewer #2: All comments have been addressed

2. Is the manuscript technically sound, and do the data support the conclusions?

Reviewer #1: Yes

Reviewer #2: Partly

3. Has the statistical analysis been performed appropriately and rigorously? 

Reviewer #1: Yes

Reviewer #2: Yes

4. Have the authors made all data underlying the findings in their manuscript fully available?

Reviewer #1: Yes

Reviewer #2: Yes

5. Is the manuscript presented in an intelligible fashion and written in standard English?

Reviewer #1: Yes

Reviewer #2: Yes

6. Review Comments to the Author

Reviewer #1: Important research work. I had reviewed a revision. Authors did a good job of incorporating the required changes.

Reviewer #2: This study’s purpose was to determine the identify a disparity in iron deficiency (ID), anemia, and iron deficiency anemia (IDA) in refugees depending on whether they had stayed in a refugee camp prior to coming to Canada. The study explains that refugee camps, oftentimes, lead to poor health outcomes and so suggested that there would be a higher prevalence of these disease among refugees who have experience such camps. However, the study finds that there is no association between refugee camp exposure and ID, anemia, or IDA. The study does conclude a relationship between ID, anemia, and IDA between gender with women had significantly more cases as well as an inverse correlation with age and months before arriving to Canada. The study does a great job of contextualizing the study in the setting of refugee camps and makes a logical relationship of refugee camps to ID, anemia, and IDA. This logical relationship builds a strong argument as to why such diseases should be studied and how refugee camps can influence the prevalence of these disease. Another strength of the paper is the logical data analysis as the study manages to use the correct statistical analyses while also highlighting the biases that may be present in the data, such as the difference in cutoff for iron deficiency depending on the statistic used. Some weaknesses of the paper relate to the study ignoring certain confounding variables such as culture with sufficient depth as well as formatting issues with regards to data presentation.

The study displays major issues with argument flow in the conclusion as well as not considering certain confounding variables. The study mentions that culture is a significant factor in disease prevalence among different ethnic groups of refugees. However, the study does not consider culture during their data analysis and does not account for biases towards certain ethnic cultures over others when investigating the impact of refugee camps. An attempt to consider culture is made on lines 184-193 but while stating the proportion of cultures is a good start, these proportions are not accounted for data analysis.

Another major issue is related to the conclusion of whether refugee camps play a role in ID, anemia, and IDA prevalence. The study concludes that it does not but does not offer any reasons as to why this may be the case. This lack of support makes the conclusion lose validity. Furthermore, the study uses its conclusions to further explain society consequences of the refugees in lines 303-305 but this extension does not seem appropriate before validating the conclusions of this study.

The limitations section of the study provides number limitations but without explanation. Explanations are important because it allows the reader to understand how the limitations may have impacted the data collection and data processing.

There were also some formatting issues in the report for data tables. Table 1 was difficult to read in some areas due to units not being clear for certain rows such as the numbers in brackets corresponding to the row “Months in Canada Prior to Blood.”

Another issue is with result presentation. A pie chart is oftentimes not suitable to display proportions exceeding two variables because the proportions become difficult to compare. Therefore, I would recommend changing the graph type to a more suitable graph where the comparisons between variables are more obvious, such as a bar graph.

Lastly, there are grammatical issues in the report. An example is the limitations paragraph which starts with a four-word topic sentence and does not adequately explain what the paragraph will be about. Furthermore, there is no flow in this paragraph as each idea is brought forward without explanation and separated by periods. There are examples of awkward syntax such as the sentence on lines 139-141 having no verb and therefore, not being a complete sentence.

Overall, the study does a great job of giving a holistic overview of ID, anemia, IDA among refugees by giving a strong foundation for the reader and also an in-depth statistical analysis of the data to come to a conclusion. However, the report has problems in addressing certain biases as well as present evidence for specific conclusions.

7. PLOS authors have the option to publish the peer review history of their article (what does this mean?). If published, this will include your full peer review and any attached files.

Reviewer #1: No

Reviewer #2: No

---

## [Author Response · Author response to Decision Letter 1]

15 Oct 2022

Marta Davidson, PhD MD FRCPC

Hematologist

Division of Medical Oncology and Hematology

Princess Margaret Cancer Center

700 University Ave, 6W091

Toronto, Ontario M5G 1Z5

October 15, 2022

To the Editor and Reviewers:

We thank you for taking the time to to review our manuscript entitled “Iron deficiency, anemia and association with refugee camp exposure among recently resettled refugees: a Canadian retrospective cohort study”, and for your insightful suggestions on how to improve it. We have addressed each comment individually below. We trust that you will find our revisions and explanations acceptable. 

Please note that the referenced pages and lines pertain to the revised manuscript in which changed have been tracked. 

1. Editor: “Please review your reference list to ensure that it is complete and correct. If you have cited papers that have been retracted, please include the rationale for doing so in the manuscript text, or remove these references and replace them with relevant current references. Any changes to the reference list should be mentioned in the rebuttal letter that accompanies your revised manuscript. If you need to cite a retracted article, indicate the article’s retracted status in the References list and also include a citation and full reference for the retraction notice.”

Response: 

The reference list has been reviewed and is complete and correct. There are no references to retracted articles.

We have updated reference #1 to include the most recent UHNCR Global Trends Report:

UNHCR. Global Trends Report: Forced Displacement in 2021: The UN Refugee Agency; 2022 [cited 2022]. Available from: https://www.unhcr.org/62a9d1494/global-trends-report-2021.

We have added 7 new recent references (Reference numbers 5, 27, 28, 42, 43, 47, 48 in the revised manuscript) to update the background information and discussion section given the time that has passed since the last submission

5. UNHCR. Shelter: The UN Refugee Agency; 2022 [Available from: https://www.unhcr.org/shelter.html.

27. Ahmed RH, Yussuf AA, Ali AA, Iyow SN, Abdulahi M, Mohamed LM, et al. Anemia among pregnant women in internally displaced camps in Mogadishu, Somalia: a cross-sectional study on prevalence, severity and associated risk factors. BMC Pregnancy Childbirth. 2021;21(1):832.

28. Engidaw MT, Wassie MM, Teferra AS. Anemia and associated factors among adolescent girls living in Aw-Barre refugee camp, Somali regional state, Southeast Ethiopia. PLoS One. 2018;13(10):e0205381.

42. Khuri J, Wang Y, Holden K, Fly AD, Mbogori T, Mueller S, et al. Dietary Intake and Nutritional Status among Refugees in Host Countries: A Systematic Review. Adv Nutr. 2022;13(5):1846-65.

43. UNHCR. World Food Programme: The UN Refugee Agency; 2022 [Available from: https://www.unhcr.org/world-food-programme-49eed2ba6.html.

47. Government of Canada. #WelcomeAfghans: Key figures: Immigration, Refugees and Citizenship Canada; 2022 [Available from: https://www.canada.ca/en/immigration-refugees-citizenship/services/refugees/afghanistan/key-figures.html.

48. Government of Canada. Canada-Ukraine Authorization for Emergency Travel: Immigration, Refugees and Citizenship Canada; 2022. Available from: https://www.canada.ca/en/immigration-refugees-citizenship/news/2022/03/canada-ukraine-authorization-for-emergency-travel.html.

2. Reviewer #1: Important research work. I had reviewed a revision. Authors did a good job of incorporating the required changes.

Response:

Thank you.

3. Reviewer #2: “…The study displays major issues with argument flow in the conclusion as well as not considering certain confounding variables. The study mentions that culture is a significant factor in disease prevalence among different ethnic groups of refugees. However, the study does not consider culture during their data analysis and does not account for biases towards certain ethnic cultures over others when investigating the impact of refugee camps. An attempt to consider culture is made on lines 184-193 but while stating the proportion of cultures is a good start, these proportions are not accounted for data analysis…”

Response:

The second paragraph of the “Statistical Analysis” portion of the “Methods” section, describes our clustering analysis that would partially address the reviewer’s concerns. (Lines: 155-163). In this analysis we found negligible clustering by UN global region of origin for any of the outcomes of interest which suggests that the associations of interest were not significantly different across global regions.:

• Page 17, Line 363-366 further addresses these concerns: “While our study is amongst the largest Canadian refugee cohort studies conducted to-date and comprised a heterogeneous global refugee cohort from 47 different countries, it is underpowered to assess country-specific variations which may factor into the lack of significant regional or sub-regional variation in our outcomes. ” 

• Line 335-350 also addresses cultures differences in the literature

4. Reviewer #2: “…Another major issue is related to the conclusion of whether refugee camps play a role in ID, anemia, and IDA prevalence. The study concludes that it does not but does not offer any reasons as to why this may be the case. This lack of support makes the conclusion lose validity…”

Response:

Thank you for pointing out our lack of clarity. This is important given our a priori motivation for this study was to investigate the association of refugee camp exposure with differences in anemia and iron deficiency outcomes among newly arrived refugees to Canada. The third paragraph of our discussion explored our null result, however, it is evident this exploration was not featured prominently nor with enough clarity. As such, we have explore our null result in more depth in paragraphs 3 and 4 of our discussion section. We have highlighted key changes as follows: 

• Line 313-333: “To our knowledge, this is the first study to investigate the association of nutritional and health status with pre-migration refugee camp exposure among global refugees, from 47 different countries, resettled to a high-income country. Overall, we found no differences in the prevalence of ID, anemia, and IDA among refugees by refugee camp exposure. Our findings suggest that refugees in non-camp settings (often in urban settings) may experience similar nutritional deficiencies as well as barriers accessing food and nutritional supports as camp-dwelling refugees. A recent systematic review also showed that nutritional deficiencies are a significant problem among non-camp dwelling refugees. (42) A number of factors may be contributing to this problem. For example, non-camp refugees may lack access to supports from the World Food Programme(43) and other international aid organizations that traditionally support refugees in camp settings. Further, the majority of displaced persons are relocated to lower income countries, (1, 42) which likely have limited resources to offer to urban-dwelling refugees. Our findings appear to contradict the purported health and economic benefits of non-camp dwelling for refugees cited by the UNHCR such as greater freedom of movement and access to employment, given that nutritional deficiencies and food access barriers are most often related to poverty.(4) 

• Line 341-350 “… These observed differences may be due to the geographic and cultural variability between different refugee camps, as well as significant differences in their availability of nutrition supports and medical services. Unlike previous studies that have been limited to specific global sub-regions and individual ethnocultural populations, our study investigated a global cohort of refugees across 47 countries of origin. We found negligible clustering by UN global region of origin for ID, anemia, and IDA, suggesting that the associations of interest were not significantly different across global regions. Together our findings suggest that ID, often related to food insecurity, is highly prevalent in pre-menopausal refugee women irrespective of pre-migration living situation or global origin.”

5. Reviewer #2: Furthermore, the study uses its conclusions to further explain society consequences of the refugees in lines 303-305 but this extension does not seem appropriate before validating the conclusions of this study.

Response:

We have revised our discussion section and conclusions as recommended to better interpret our study findings in the context of our a priori hypothesis. We then clarify our considerations around the potential implications of our study findings to the 98% of refugees who are not resettled to high income countries such as Canada and thus not able to undergo similar, laboratory-based health characterizations. We revised our manuscript as follows:

• Line 313-333: “To our knowledge, this is the first study to investigate the association of nutritional and health status with pre-migration refugee camp exposure among global refugees, from 47 different countries, resettled to a high-income country. Overall, we found no differences in the prevalence of ID, anemia, and IDA among refugees by refugee camp exposure. Our findings suggest that refugees in non-camp settings (often in urban settings) may experience similar nutritional deficiencies as well as barriers accessing food and nutritional supports as camp-dwelling refugees. A recent systematic review also showed that nutritional deficiencies are a significant problem among non-camp dwelling refugees. (42) A number of factors may be contributing to this problem. For example, non-camp refugees may lack access to supports from the World Food Programme(43) and other international aid organizations that traditionally support refugees in camp settings. Further, the majority of displaced persons are relocated to lower income countries, (1, 42) which likely have limited resources to offer to urban-dwelling refugees. Our findings appear to contradict the purported health and economic benefits of non-camp dwelling for refugees cited by the UNHCR such as greater freedom of movement and access to employment, given that nutritional deficiencies and food access barriers are most often related to poverty.(4) 

• Line 398-406: Globally, less than 2% of refugees are resettled in high income countries such as Canada; however, studies among multinational cohorts of recently resettled refugees can improve understanding of health conditions faced by refugees regionally and universally. (50) Future studies of ID and anemia among global refugees would help to elucidate if regional and cultural differences exist, and if their prevalence varies across different nutritional and medical assistance models between refugee camps. Improved understanding of nutritional deficiencies in refugee populations may inform treatment and prevention programs, such as targeted iron supplementation programs for pre-menopausal refugee women and adolescent girls. 

6. Reviewer #2: The limitations section of the study provides number limitations but without explanation. Explanations are important because it allows the reader to understand how the limitations may have impacted the data collection and data processing.

Response:

Thank you for pointing out our lack of clarity. We were limited by word count in our initial submission but have revised our limitations paragraph and elaborated on each point to explain how each limitation pertains to our study results and possibly impacts on conclusions. Our revisions are as follows: 

• Line 358-379: Our study has limitations that warrant consideration. As our study was conducted at a single site, it may not be representative of refugee populations in other host-regions. Further, our study enrolment ended prior to the influx of approximately 41,000 Syrian refugees to Canada in 2015-2016, and more recent Afghan and Ukrainian refugees(46, 47), thus, it may not represent the current global refugee population. While our study is amongst the largest Canadian refugee cohort studies conducted to-date and comprised a heterogeneous global refugee cohort from 47 different countries, it is underpowered to assess country-specific variations which may factor into the lack of significant regional or sub-regional variation in our outcomes. Moreover, we only identified two cases of severe anemia in our study and may underestimate the prevalence of severe anemia cases that may present to acute care settings. This may be due to a selection bias given that our study relied on outpatient laboratory tests. Alternatively, selection pressures during the refugee migration process may select against individuals with severe anemia who may succumb to its complications or be unable to travel.(48) In addition, because our study was restricted to adult refugees and claimants, post-menarche adolescent girls, who likely also experience high prevalence of ID, anemia, and IDA were not included, thus potentially underestimating their prevalence. Finally, in our cohort, 19% of refugees had resided in a refugee camp prior to arrival in Canada, lower than current global trends, where 24% of global refugees reside in camps,(3) suggesting camp-exposure may be underrepresented in our cohort. 

7. Reviewer #2: “There were also some formatting issues in the report for data tables. Table 1 was difficult to read in some areas due to units not being clear for certain rows such as the numbers in brackets corresponding to the row “Months in Canada Prior to Blood.”

Response:

Thank you for pointing this out. We have now corrected the formatting of Table 1 so that the alignment is improved and it is easier to follow. Our revised Table 1 can be found on p.10-11 line 203-207 in the manuscript and below:

Table 1. Patient characteristics by sex and refugee camp exposure (N = 1032).

Variable Female N = 534 Male N = 498

 Refugee Camp +

N = 94 Refugee Camp -

N = 440 p-value* Refugee Camp +

N = 99 Refugee Camp -

N = 399 p-value*

Age (years) – Median 

[IQR] 30.3

[23.6-38.3] 31.4

[25.5-40.0] 0.28 32.0

[25.5-42.4] 32.7

[26.5-41.1] 0.92

Months in Canada Prior to Blood Tests – Median 

[IQR] 

1.1

[0.7-1.9] 1.4

[0.8-2.5] 0.08 1.2

[0.8-2.1] 1.3

[0.7-2.6] 0.69

UN Global Region – N (%) 

 Africa 63 (67.0) 206 (46.8) < 0.01 69 (69.7) 204 (51.1) < 0.01

 Americas 0 (0) 17 (3.9) 0 (0) 5 (1.3) 

 Asia 31 (33) 210 (47.7) 30 (30.3) 190 (47.6) 

 Europe 0 (0) 7 (1.6) 0 (0) 0 (0) 

Number of Children – 

Median 

[IQR] 1

[0-4] 2

[0-3] 0.43 1

[0-2] 1

[0-2] 0.86

Pregnancy^ – N (%) 

 Yes 6 (6.4) 48 (10.9) 0.19 N/A 

 No 88 (93.6) 392 (89.1) 

Refugee Category# 

 GAR 71 (75.5) 204 (46.4) < 0.01 65 (65.7) 188 (47.1) < 0.01

 PSR 23 (24.5) 191 (43.4) 34 (34.3) 180 (45.1) 

 Claimant 0 (0) 45 (10.2) 0 (0) 31 (7.8) 

English Language proficiency at intake– N (%) 

 None 61 (64.9) 214 (48.6) 38 (38.4) 129 (32.3) 

 Limited 11 (11.7) 73 (16.6) 20 (20.2) 86 (21.6) 

 Good 22 (23.4) 153 (34.8) 0.02 41 (41.4) 184 (46.1) 0.52

*p-values calculated using a Wilcoxon rank sum test for comparison of medians and chi-square or Fisher’s exact tests for comparison of frequencies. #GAR refer to Government assisted refugees, PSR refers to privately sponsored refugees, and Claimant refers to refugee claimants or asylum seeker. ^Pregnant refers to active pregnancy at the time of laboratory investigations. N/A refers to not applicable.

8. Reviewer 2: “Another issue is with result presentation. A pie chart is oftentimes not suitable to display proportions exceeding two variables because the proportions become difficult to compare. Therefore, I would recommend changing the graph type to a more suitable graph where the comparisons between variables are more obvious, such as a bar graph.”

Response:

Thank you for pointing this out. We have now changed our pie chart (Figure 4) to a bar graph as follows:

9. Reviewer #2: Lastly, there are grammatical issues in the report. An example is the limitations paragraph which starts with a four-word topic sentence and does not adequately explain what the paragraph will be about. Furthermore, there is no flow in this paragraph as each idea is brought forward without explanation and separated by periods.

Response:

Thank you for pointing out our lack of clarity. We have extensively revised our limitation paragraph to improve its coherence and logical flow. 

10. Reviewer #2: There are examples of awkward syntax such as the sentence on lines 139-141 having no verb and therefore, not being a complete sentence.

Response:

We have revised the manuscript throughout to improve grammar, flow, and coherence. Our grammatical revisions are highlighted in the manuscript with tracked changes. 

In addition, we grammatically revised the paragraph noted above on lines 139-141 (lines 122-127 in revised manuscript). Our revisions are as follows:

• p.6 122-127: “We defined mild, moderate, and severe anemia among non-pregnant women as a hemoglobin concentration of: 110-119g/L, 80-109g/L, and <80g/L, respectively, and among pregnant women as 100-109g/L, 70-99g/L, and <70g/L, respectively.”

We thank the editor and reviwers for the opportunity to revise our manuscript and hope that you will find our revisions satisfactory.

Sincerely,

Marta Davidson MD, PhD, FRCPC

Hematologist

Princess Margaret Hospital

Toronto, Ontario

---

## [Editor Report · Decision Letter 2]

28 Nov 2022

Iron deficiency, Anemia and Association with Refugee Camp Exposure Among Recently Resettled Refugees:  a Canadian retrospective cohort study

PONE-D-21-12986R2

Dear Dr. Davidson,

We’re pleased to inform you that your manuscript has been judged scientifically suitable for publication and will be formally accepted for publication once it meets all outstanding technical requirements.

Kind regards,

Mabel Aoun, MD, MPH

Academic Editor

PLOS ONE
---

## [Editor Report · Acceptance letter]

6 Dec 2022

PONE-D-21-12986R2 

Iron deficiency, anemia and association with refugee camp exposure among recently resettled refugees: a Canadian retrospective cohort study 

Dear Dr. Davidson:

I'm pleased to inform you that your manuscript has been deemed suitable for publication in PLOS ONE. Congratulations! Your manuscript is now with our production department. 

Kind regards, 

on behalf of

Dr. Mabel Aoun 

Academic Editor

PLOS ONE